# PARP Inhibitors in Brain Metastases from Epithelial Ovarian Cancer through a Multimodal Patient Journey: Case Reports and Literature Review

**DOI:** 10.3390/ijms25147887

**Published:** 2024-07-18

**Authors:** Simona Frezzini, Giulia Tasca, Lucia Borgato, Lucia Sartor, Annamaria Ferrero, Grazia Artioli, Alessandra Modena, Alessandra Baldoni

**Affiliations:** 1Medical Oncology 3 Unit, Veneto Institute of Oncology IOV—IRCCS, 35128 Padova, Italy; 2Medical Oncology 2 Unit, Veneto Institute of Oncology IOV—IRCCS, 35128 Padova, Italy; giulia.tasca@iov.veneto.it; 3Medical Oncology Unit, Ospedale San Bortolo, 36100 Vicenza, Italy; lucia.borgato@aulss8.veneto.it; 4Medical Oncology Unit, Ospedale di Camposampiero, ULSS 6 Euganea, 35131 Padova, Italy; lucia.sartor@aulss6.veneto.it; 5Academic Department Gynecologic Oncology, Mauriziano Hospital, University of Torino, 10124 Torino, Italy; annamaria.ferrero@unito.it; 6Medical Oncology Unit, Azienda ULSS 2 Marca Trevigiana, 31100 Treviso, Italy; grazia.artioli@aulss2.veneto.it; 7Medical Oncology Department, IRCCS Ospedale Sacro Cuore Don Calabria, 37024 Negrar, Italy; alessandra.modena@sacrocuore.it; 8Oncology and Hematology Department, Mirano AULSS3 Serenissima, 30035 Mirano, Italy; alessandra.baldoni@aulss3.veneto.it

**Keywords:** epithelial ovarian cancer, brain metastases, PARP inhibitors, multimodal patient journey

## Abstract

Epithelial ovarian cancer (EOC) is the deadliest gynecological malignancy worldwide. Brain metastasis (BM) is quite an uncommon presentation. However, the likelihood of central nervous system (CNS) metastasization should be considered in the context of disseminated disease. The therapeutic management of BMs is an unmet clinical need, to date. We identified, across different cancer centers, six cases of both BRCA wild-type and BRCA-mutated EOCs spreading to the CNS. They presented either with a single brain lesion or with multiple lesions and most of them had intracranial-only disease. All cases received Poly-ADP ribose polymerase inhibitor (PARPi) maintenance, as per clinical practice, for a long time within a multimodal treatment approach. We also provide an insight into the available body of work regarding the management of this intriguing disease setting, with a glimpse of future therapeutic challenges. Despite the lack of unanimous guidelines, multimodal care pathways should be encouraged for the optimal disease control of this unfortunate patient subset. Albeit not being directly investigated in BM patients, PARPi maintenance is deemed to have a valuable role in this setting. Prospective research, aimed to implement worthwhile strategies in the multimodal patient journey of BMs from EOC, is eagerly awaited.

## 1. Introduction

### 1.1. Clinical Presentation of BMs from EOC

EOC, also referred to as the “silent killer” or “whispering disease” [1], harbors the highest disease burden and mortality among gynecologic malignancies, due to earlier relapse despite optimal chemosensitivity [1,2,3]. Regarding metastatic patterns, the locoregional relapse, caused by intraperitoneal or lymphatic spread, is highly frequent within 3 years from adjuvant platinum-based therapy completion [4]. The abdominal–pelvic metastatic sites are more common than hematogenous distant sites [4]. Among them, CNS is exceedingly rare in EOC, being involved in about 0.3–11% of different series [5]. However, better clinician awareness and the latest advances in anticancer therapies as well as imaging techniques have led to an augmented incidence of unusual CNS metastases [5,6].

Evidence regarding this patient subset is still sparse and controversial [7]. BM is an unfavorable event with a very poor prognosis [7]. CNS spreading in EOC patients is also a late clinical manifestation, with a time to onset of BM significantly longer than that for other recurrent sites. Strikingly, high-grade serous ovarian cancer (HGSOC) was found as the most common histotype related to BM development, with a longer time interval despite more aggressive behavior [7,8]. The clinical presentation of BM includes both oligometastatic and polymetastatic lesions [9], although most patients harbor a single metastasis [10]. The most typical clinical complaints are sensory and motor disturbances, cognitive dysfunction, or intracranial hypertension symptoms [7,11]. Historically, EOC, Karnofsky performance status (KPS) > 70, single brain metastasis, absence of extracranial disease, cranial surgery, cranial RT, and chemotherapy (CHT) have been found as independent favorable predictors of overall survival (OS) in a multivariate analysis [9]. Recently, younger age, intracranial-only disease, single CNS site, and multimodality care were established as good prognostic indicators for longer OS after BM diagnosis [7].

### 1.2. The Key Role of the Multimodal Treatment Plan for BMs from EOC

Due to the high disease burden and the heterogeneous uptake of anticancer drugs, an effective treatment of BMs is a serious unmet clinical need in the field of neuro-oncology [5,12]. Collectively, no consensus exists about the optimal treatment strategy for EOC BMs due to their rarity [7]. Only retrospective data support clinical decision making in this subset [5]. An aggressive multimodal approach is warranted in carefully selected patients who may profit from actual intracranial disease control. The therapeutic mainstay of BM treatment consists of surgical resection, whole-brain radiotherapy (WBRT), CHT, and, more recently, intensity-modulated RT (IMRT) and stereotactic radiosurgery (SRS), including gamma-knife radiosurgery [5,13]. The new techniques of CNS radiation (IMRT and SRS) should be prioritized for tumor control due to the long-term sequelae of WBRT, unless intracranial disease is widespread [5,14]. In line with this notion, SRS yields more favorable outcomes in terms of OS, neurological impairment, and QoL in highly selected patient cohorts (oligometastases, good disease control, and good KPS) [5]. A good KPS, a single disease site, and good tumor accessibility are the main requirements for cranial surgery, also aimed at histological confirmation [5,15]. In the largest single-institution study of EOC BMs, postoperative RT ensured better intracranial disease control as compared to surgery alone [9]. In lieu of current findings, the highest median OS was reported for BM patients receiving all therapeutic modalities including CHT [10,11].

### 1.3. Biomarkers of CNS Spread and Potential Therapeutic Targets

Presently, the evidence for potential biomarkers of brain spread from EOC is almost exclusively based on small-sized retrospective studies, while prospective validation is lacking [4,16,17]. The standard platinum-based regimens are the most used due to their ability to cross the blood–brain barrier (BBB), with beneficial effects based on the known platinum sensitivity of EOC. The long-term prognosis is still unsatisfactory [18]. Scant data suggest the possibly of the predictive role of hormone receptors as well as the multi drug reactivity 1 gene expression as possible biomarkers of BM development, but further prospective research clarifying this finding is of great interest [4,19].

Both BRCA1 and BRCA2 tumor suppressor genes play a key role in high-fidelity DNA repair via the homologous recombination repair (HRR) pathway. A dysfunctional HRR, which is also referred to as the homologous recombination deficiency (HRD) signature, is broadly identified in about half of HGSOC patients [19]. All the genetic/epigenetic defects (not limited to BRCA mutations) included in the HRD signature identify somatic mutational landscapes reflecting the BRCAness phenotype. The HRD signature may serve as a biomarker for platinum and PARPi sensitivity [20], and thus, may inform prognosis and treatment decision making of BRCA-like tumors, translating into longer survival time and time to platinum-resistance [21]. Strikingly, the presence of BRCA pathogenic mutations has been currently suggested as a risk factor for brain spread from EOC [4]. BRCA1-2-mutant patients are prone to develop earlier extraperitoneal visceral metastases, such as BMs, as opposed to their wild-type counterparts [22,23], with a shorter median time to metastasis and a younger age without survival differences by BRCA status [23]. A limited cancer burden was reported in BRCA-related BMs, partly due to known platinum sensitivity enabling durable systemic disease control [16]. Consistently, the improved survival due to the known platinum sensitivity of BRCA-mutant patients could also favor the higher incidence of uncommon CNS metastases [16].

To date, the functional role of BRCA deficiency in EOC brain metastasization remains unexplained [23]. Likewise, the evidence suggesting HRD as a likely risk factor for BM [15] may explain the similar survival trend in both the BRCA wild-type and BRCA-mutant subgroups [5]. In BRCA wild-type cases, the presence of at least one mutation in alternative HRD genes acts as a risk factor for BM development thus supporting PARPi use in this population with BMs [5].

Therefore, the data supporting the implication of BRCA and the HRD signature in EOC spreading to the brain, while inconclusive, are noteworthy. The first NGS study exploring the set of actionable somatic mutations in metastatic EOC found a high number of BRCA1/2 mutations in addition to other HRR defects in all sequenced BM samples [24]. These findings strongly suggest that pharmacological PARP inhibition could be an attractive targeted therapeutic for patients with BMs [24]. To date, PARPi maintenance is thought to have a valuable role in the management of this patient subset [5]. Given the paucity of EOC patients affected by BMs, no unanimous guidelines are yet in sight [7]; thereby, the therapeutic algorithm in this setting needs to be clarified [15].

Herein, we present a multi-institutional case study research highlighting the clinical outcomes of EOC patients with intracranial disease who received PARPi maintenance during their therapeutic journey (as summarized in Table 1, below).

## 2. Case Presentation

### 2.1. Case One

A 53-year-old postmenopausal female with an unremarkable personal medical history was diagnosed in September 2019 with advanced ovarian cancer with widespread peritoneal carcinomatosis along with bilateral pleural effusion. After diagnostic and staging laparoscopy, she underwent primary debulking surgery with optimal cytoreduction with FIGO IIIC Surgical Staging of HGSOC on histopathological exam. The baseline CA125 was 1133 U/mL. Following this, from November 2019 to March 2020, she was offered frontline 3-weekly Carboplatin plus Paclitaxel plus Bevacizumab (6 cycles) followed by 3-weekly Bevacizumab maintenance monotherapy for up to 22 cycles. During the maintenance phase, the BRCA germline testing reported a pathogenic BRCA1 deleterious mutation.

In October 2021, the patient experienced limited intracranial relapse with a single cerebellar nodule in the absence of clinical complaints. A concomitant rising of CA125 (183 U/mL) occurred. Hence, after multidisciplinary agreement, she was deemed suitable for suboccipital craniotomy with radical resection of the lesion; pathology highlighted “brain metastases from HGSOC”. Postoperatively, CA125 was normalized (18 U/mL). In December 2021, the restaging CT scan revealed an enlarged para-aortic lymph node (minimum axis of 15 mm) without other distant sites; thus, a second-line treatment with the Carboplatin plus Pegylated liposomal doxorubicin (PLD) doublet was delivered for 6 courses, followed by a complete radiological response of the target lymphadenopathy. Contemporarily, the findings on follow-up brain MRI were more suggestive of a likely residue at the surgical bed as compared to a local relapse.

Due to the diagnostic challenge of brain metastasis concomitantly with extracranial remission, after CHT completion, in June 2022, local SRS targeting all the posterior cranial fossa to a dose of 30 Gy in 10 fractions was performed. Due to the intracranial-only tumor burden, despite BRCA mutation, the multidisciplinary tumor board (MTB) at the referral institute deemed the patient a candidate for Niraparib maintenance monotherapy, which started in August 2022 and is currently ongoing. Over time, reassessment brain MRIs have shown no signs of intracranial relapse. No dose-limiting toxicity has been reported at present. The CA125 marker has remained within the normal range. All the patient journey of case one is depicted in Figure 1, below.

### 2.2. Case Two

A 52-year-old postmenopausal female with an unremarkable family history was diagnosed in April 2014 with advanced EOC (diagnostic laparoscopy). At the referral institute, due to disease burden, she underwent perioperative chemotherapy with a 3-weekly Carboplatin plus Paclitaxel regimen (three courses) followed by interval debulking surgery in July 2014 with optimal cytoreduction (no gross residual disease). Pathology confirmed FIGO IIIC HGSOC. In the adjuvant phase, from September 2014 to January 2015, the patient received a further six cycles of the same platinum-based regimen combined with 3-weekly Bevacizumab at a dose of 15 mg/kg. Then Bevacizumab maintenance monotherapy was delivered until January 2016. Meanwhile, in November 2015, the BRCA germline testing was remarkable for a BRCA1 pathogenic variant.

In May 2018, CNS recurrence occurred with diagnostic work-up revealing only a right parietal lesion (of 15 mm axis); the systemic CT scan excluded extracranial relapse. Following MTB discussion, intracranial SRS (27 Gy/3 fractions) was performed, followed by 6 courses of Carboplatin AUC 6 monotherapy, with overall good tolerance. In December 2018, owing to platinum-sensitive relapse, despite mutational status, Niraparib maintenance was started at a dose of 200 mg once daily, based on baseline body weight and platelet count. In September 2023, the follow-up brain MRI was suggestive for complete remission of brain metastases and the repeat PET-CT scan was unremarkable for systemic disease. Concomitantly, tumor markers were unremarkable; thus a 6-month radiologic reassessment was established. The Niraparib maintenance is currently ongoing with good subjective tolerance and no relevant toxicities. Therefore, the patient is experiencing a durable clinical benefit throughout Niraparib therapy. All the patient journey of case two is depicted in Figure 2, below.

### 2.3. Case Three

A 47-year-old postmenopausal female with an unremarkable medical history was diagnosed in March 2007 with advanced fallopian tube (FT) cancer. At the referral institute she underwent primary debulking surgery with optimal cytoreduction (complete response intraoperatively); pathology confirmed FIGO IIIB HGSOC. Following that, she was given the frontline Carboplatin plus Paclitaxel regimen (6 cycles), which completed in September 2007. Follow-up was negative until November 2012, when a CT scan revealed a pelvic mass infiltrating the sigmoid colon, which was deemed suitable for surgical resection (sigmoidectomy along with colorectal anastomosis). Once bowel metastasis from HGSOC was histologically confirmed, the patient was offered the Carboplatin plus PLD doublet, and then followed up. Meanwhile, in May 2013, the patient tested positive for germline BRCA1 deleterious mutation. Then, she experienced pulmonary and mediastinal relapse, not believed to be suitable for locoregional approaches. Thereby, she was given Carboplatin AUC4 monotherapy (6 cycles), with partial radiological response of the disease sites.

The patient remained free of disease progression until January 2016, when she complained of positional headaches, dizziness, and blurred vision. Intracranial recurrence due to temporo-parietal and occipital lesions concomitantly with stable extracranial disease was reported on the restaging CT scan. After multidisciplinary agreement, intracranial SRS was performed, followed by 6 courses of the Carboplatin–Paclitaxel doublet, yielding partial remission on all disease sites.

In August 2016, considering platinum-sensitive relapse, Olaparib maintenance was started. She experienced 21-month disease control throughout Olaparib therapy. Following new pulmonary progression in May 2018, a further Carboplatin-based doublet was delivered for 6 cycles, yielding intra- and extracranial stable disease up to April 2019. Afterwards, due to systemic progression at cerebellum and supra-/infra-diaphragmatic lymph nodes, the patient underwent a further three chemotherapeutics, namely the PLD-trabectedin doublet, Carboplatin, and weekly Paclitaxel. Thereafter, the patient died in November 2019 after being hospitalized for pulmonary distress. All the patient journey of case one is depicted in Figure 3, below.

### 2.4. Case Four

A 65-year-old postmenopausal female without comorbidities sought medical attention in April 2016 for abdominal swelling and pain, with radiological assessment of omental cake, diffuse peritoneal carcinomatosis, ascites, and infradiaphragmatic lymphadenopathies. Due to the disease burden and the histological diagnosis of HGSOC on laparoscopic biopsies, at the referral center, the patient received a neoadjuvant Carboplatin plus Paclitaxel regimen, with good radiologic response. In June 2016, she underwent interval debulking surgery comprising bilateral salpingo-adnexectomy, omentectomy, appendectomy, peritonectomy and pelvic/para-aortic lymphadenectomy, with no gross residual disease. Pathology highlighted FIGO IIIC HGSOC. Meanwhile, BRCA germline testing excluded a pathogenic mutation. The subsequent follow-up remained negative until March 2019, when relapse occurred. The CT findings were the peritoneal implants of Glisson’s capsule and pelvic peritoneum, in the context of mild pain. Owing to the fully platinum-sensitive relapse, the patient was offered a 2nd line Carboplatin (then switched to Cisplatin due to allergy) plus Gemcitabine regimen with a complete radiological response of all disease sites. Thus, in June 2019, Niraparib maintenance monotherapy was started at the full dose of 300 mg once daily with weekly full blood count testing as per the drug label. Due to recurrent grade 2 thrombocytopenia in the first 3 cycles leading to a 3-week break, Niraparib was de-escalated to 200 mg once daily without adjustments until the treatment ended.

In May 2020, neurological complaints (fasting emesis, headaches, and unsteady gait) occurred. Intracranial disease was confirmed on brain MRI due to a single left cerebellar metastasis of 26 × 30 mm with modest vasogenic edema, along with no extracranial disease on CT scan. Following multidisciplinary agreement, the patient, after short course of anti-edema corticosteroids, underwent suboccipital craniotomy with radical resection of the single cerebellar lesion; pathology highlighted “brain metastases from HGSOC”. A postsurgical sequela (pseudomeningocele) required immediate surgical revision.

The neurological interdisciplinary care group at the referral center, based on the time elapsed from neurosurgery, did not recommend adjuvant RT on surgical bed. Thus, Niraparib was resumed after surgical wound healing, given the extracranial disease control along with the low-volume intracranial disease. In November 2020, the follow-up brain MRI showed findings more suggestive for a deep residue of the resected metastasis than a local relapse, thus posing a diagnostic dilemma. Due to the clinical complaints (unsteady gait), a diagnostic lumbar puncture was performed in an inpatient setting, excluding leptomeningeal metastases (LMs), followed by local SRS targeting all the posterior cranial fossa (25 Gy/5 fractions). Meanwhile, Niraparib was maintained, aside from a 5-day break during RT sessions. The patient experienced a sustained clinical benefit from Niraparib monotherapy with improved kinesthesia by virtue of daily functional rehabilitation and no new neurological complaints. Overall intra- and extracranial disease control on follow-up brain MRI and CT scans, respectively, was shown.

In December 2020, she was hospitalized due to intracranial hypertension symptomatology with subsequent clinical deterioration. MRI findings were suggestive of LMs, proven by cerebrospinal fluid cytology. A systemic CT scan excluded extracranial progression. Due to poor KPS and a rapidly progressing disease, the MTB at the referral center retained the patient as unfit for CHT rechallenge. Supportive and palliative care with intensive rehabilitation were encouraged upon discharge with clinical benefits to symptom relief. At the end of May 2021, 12 months after BM diagnosis, the patient died. Hence, a multimodal care approach combining surgery, RT, and CHT has been encouraged to manage brain metastases from EOC in a fit patient with good KPS. All the patient journey of case one is depicted in Figure 4, below.

### 2.5. Case Five

A 52-year-old postmenopausal female with an unremarkable medical history was diagnosed in May 2019 with advanced EOC. At the referral institute she underwent primary debulking surgery with optimal cytoreduction (no gross residual disease); pathology confirmed FIGO IIB high-grade endometrioid ovarian carcinoma. Then, she underwent a frontline Carboplatin plus Paclitaxel regimen (4 cycles out of 6, due to the patient’s intolerance). In April 2020, the BRCA germline testing (delayed due to patient’s unwillingness) resulted in a positive result for BRCA1 deleterious mutation.

Follow-up was negative until January 2021, when a lymph node relapse was detected on the restaging CT scan (para-aortic adenopathies), concomitantly with a biochemical relapse, albeit without clinical symptoms. After multidisciplinary agreement, in February 2021, the patient underwent para-aortic lymphadenectomy, with histological confirmation of “metastasis from endometrioid histotype of ovarian carcinoma”. Therefore, she was offered the Carboplatin plus PLD doublet, due to residual taxane-related neurotoxicity, followed by Olaparib maintenance, which was started in October 2021. Olaparib was de-escalated (to 450 mg daily), due to anemia, without further adjustments subsequently.

In December 2022, she was hospitalized owing to an epileptic crisis without other neurological complaints, and a CT scan revealed a single left parieto-occipital lesion of about 43 mm without extracranial disease sites. Due to intracranial-only disease, the MTB at the referral center deemed the patient a candidate for neurosurgery with radical resection of the single lesion and postoperative SRS (3 fractions). Thus, the patient was referred for Olaparib maintenance resumption (apart from 3 weeks off during the treatments). A histological report of “brain metastases from G3 endometrioid ovarian cancer” was obtained.

As of June 2024, the Olaparib maintenance is currently underway, with a 32-month disease control as the last CT scan was unremarkable for intra- and extracranial disease along with a CA125 within the normal range. Thus, the patient is experiencing a long-term clinical benefit throughout the Olaparib monotherapy, without any relevant toxicity or clinical complaint. All the patient journey of case one is depicted in Figure 5, below.

### 2.6. Case Six

A 73-year-old female with an unremarkable family history sought medical attention in December 2021 for abdominal swelling and severe constipation. Diagnostic work-up revealed an adnexal mass, ascites, peritoneal carcinomatosis, omental cake, and multiple mesenteric lymphadenopathies. Baseline CA125 was 788 U/mL. At the referral center, in February 2022, she underwent laparoscopy with a histological diagnosis of HGSOC from an omental biopsy. After MTB discussion, due to the disease burden, the patient was offered neoadjuvant chemotherapy with four cycles of the Carboplatin plus Paclitaxel doublet, gaining a favorable radiological response and complete biochemical remission (CA125: 29 U/mL). In July 2022, she underwent interval debulking surgery, including retrograde hysterectomy, salpingo-oophorectomy, radical omentectomy, and excision of bulky and para-aortic, paracaval, and iliac-obturator lymph nodes, with no gross residual disease. Final pathology confirmed FIGO IIIA (ypT3a ypN1b) HGSOC. Additionally, the left obturator lymph nodes showed a localization of chronic lymphocytic leukemia/small lymphocytic lymphoma (CLL/SLL, based on WHO 2017). After hematological consultation recommending only follow-up (with no systemic therapy) for the newly diagnosed CLL/SLL, the patient received the last two cycles of platinum chemotherapy up to September 2022. Meanwhile, the HRD and BRCA somatic testing reported a BRCA wild-type and HR proficient status. Owing to the good radiological and biochemical response, based on mutational status, Niraparib maintenance monotherapy was started in October 2022 at a dose of 200 mg once daily. The first 6-month follow-up CT scan showed no evidence of disease recurrence with a concomitant increase in the number and size of pelvic lymph nodes, corresponding to the disease sites of known CLL/SLL. However, the PET-CT scan was unremarkable, such that no specific treatment was indicated by the consultant hematologist. The patient has continued Niraparib maintenance without significant toxicity.

In August 2023, she presented to the Emergency Department with worsening vomiting, nausea, and vertigo. A brain CT highlighted a large hypodense area of 43 × 21 mm in the left cerebellar hemisphere causing a mass effect on the vermis with displacement to the right and marked compression of the ventricular system. These findings were confirmed on brain MRI and were suggestive of CNS metastases. The restaging CT scan excluded extracranial relapse while confirming the known lymphadenopathies as unchanged. After transfer to the neurosurgery department, the patient underwent excision of the left cerebellar lesion. Histopathology highlighted “poorly differentiated adenocarcinoma with immunophenotypic profile consistent with the primary HGSOC, with clear resection margins”. The MTB at the referral center retained the patient as suitable for SRS to the resection bed and Niraparib resumption (after a break during neurosurgery) under close clinical and radiological surveillance.

However, about a month later, the patient returned to the Emergency Department with spontaneous hematomas and atraumatic conjunctival and eyelid hemorrhages, revealing pancytopenia (hemoglobin 7.41 g/dL, neutrophils 1.40 × 10^3^/μL, platelets 6.77 × 10^3^/μL). Niraparib was discontinued, and a bone marrow biopsy showed hypocellular marrow with CLL/SLL infiltration (25%). After normalization of blood parameters within 28 days, Niraparib was resumed with first-level dose de-escalation at 100 mg once daily. The subsequent blood count tests showed normal hematological values, thus allowing Niraparib continuation. A follow-up systemic CT scan performed in December 2023 revealed stable lymph nodes with no signs of progression. As of June 2024, the patient is asymptomatic and is continuing Niraparib maintenance therapy without any toxicity. All the patient journey of case one is depicted in Figure 6, below.

## 3. Discussion and Literature Review

In our multi-institutional case study research, six patients presenting with BMs from EOC experienced a sustained clinical benefit from PARPi maintenance received in the context of a multimodal treatment journey. Little data are available, to date, about the treatment modalities for this intriguing disease setting, due to the rarity of BMs [15]. Findings supporting PARPi use in this context are still anecdotal [5,15]. Our descriptive research can thus provide a valuable reference to clinical practice for this uncommon scenario. The current body of work about PARPi use in BMs is summarized in the Appendix A, along with the search criteria reported in the *search strategy section* (at the bottom of Appendix A).

### 3.1. What Is the Rationale behind PARPi Use in BMs from EOC?

PARPi is the standard-of-care maintenance treatment licensed both in frontline and platinum-sensitive relapse settings, even regardless of BRCA and HRD mutation status [25]. Historically, the milestone of the PARPi mechanism of action has been identified as “synthetic lethality”, consisting in the loss-of-function mutation of BRCA genes coupled with synthetically inhibiting PARP1 [21]. Recently, the key role of stalled replication forks due to PARP1 blocking, which enables genomic instability and cell death, has been suggested [21].

Limited evidence supports PARPi effectiveness in BM treatment for EOC [14,15,26]. The BBB disruption in the context of intracranial metastases could enable more successful delivery and efficacy of cytotoxics, like platinum salts and PARPi [14]. Preclinical findings strongly support the differential pharmacokinetic (PK) profile and antitumor activity of PARPi in intracranial xenografts [14]. A comparative PK study of PARPi highlighted the correlation between Niraparib’s favorable pharmacokinetic properties and preclinical antitumor effects in BRCA wild-type tumors [27,28]. Niraparib exhibits greater and sustained intratumoral exposure than Olaparib, due to higher permeability across the intact BBB [4,28]. This is consistent with the more potent tumor growth inhibition exerted by Niraparib in BRCA-mutant intracranial xenografts, as opposed to Olaparib, thus supporting its broader clinical effect in patients with both BRCAmut and BRCA wild-type tumors.

In the rapidly evolving therapeutic landscape of EOC, PARPi is increasingly employed as the maintenance monotherapy in BM patients, although its efficacy in platinum-sensitive brain relapse is almost completely unknown [18]. The unique body of work supporting PARPi use in this setting arises from review case series and retrospective studies, in the absence of robust prospective data [5].

Interestingly, broadening the therapeutic applicability of PARPis to cancers with the BRCAness phenotype, including many CNS malignancies, remains a significant challenge and an active research topic [29]. Thanks to recent preclinical studies, novel sensitivity biomarkers of BRCAness are being discovered, paving ways towards rational combinations of PARPis in neuro-oncology [29], whose feasibility and efficacy will be also informed by the ongoing clinical trials (ClinicalTrials.gov identifier: NCT03991832) [29,30]. Notably, Niraparib activity and efficacy are being tested in investigational trials for either newly diagnosed or recurrent glioblastoma patients (ClinicalTrials.gov identifier: NCT05076513 and NCT04715620) [30,31]. To the best of our knowledge, no trials on PARPi efficacy in BM patients are underway. Anyway, PARPi maintenance is gaining a growing role in the management of BMs from EOC [5].

### 3.2. What Are the Main Highlights for Niraparib Maintenance in BMs from EOC?

Cases one and two are emblematic for Niraparib maintenance in patients with intracranial-only recurrence presenting with single cerebellar and parietal oligometastases, respectively. However, case one also had para-aortic metastasis very near (2 months) the time of the BM. From an anatomical standpoint, the cerebellum represents the most common intracranial metastatic site [9]. There is a paucity of data regarding the treatment of brain metastases with Niraparib in EOC. Among the handful of available reports [15,18,26], in the former, a BRCA1-mutant patient developed a solitary fully platinum-sensitive CNS relapse from HGSOC that, nevertheless, was rapidly progressive while on platinum CHT and WBRT. After a partial response to the following platinum-based regimen, maintenance Niraparib favored intracranial stable disease and systemic control, leading to symptom relief and durable patient remission for over 17 months [26] (as reported in Appendix A).

In a different situation, a patient with CNS relapse received Niraparib monotherapy, in the ambit of a multimodal treatment plan (with WBRT and CHT), yielding an almost complete radiological response (on MRI) of brain lesions after 9 months of administration. The tolerability profile was manageable with transient asymptomatic myelosuppression favoring treatment resumption and continuation, without further events [18]. Strikingly, a very atypical presentation of a cerebellar metastasis from EOC at initial diagnosis has been recently reported [10]. A personalized multimodal therapeutic approach including neurosurgery, frontline CHT, and Niraparib maintenance led to intracranial complete clinical response and good patient quality of life [10] (as detailed in Appendix A). Notably, in a more recent retrospective cohort, BM patients receiving Niraparib as part of their multimodal treatment plan gained good disease control irrespective of their BRCA mutational status [5] (as detailed in Appendix A). In another case study, the administration of Niraparib maintenance monotherapy, following surgical debulking of a single brain metastasis, yielded long-term clinical benefit (PFS of about 29 months) in a patient who was unfit for other systemic and locoregional approaches (CHT and brain RT, respectively) [32]. Strikingly, the successful outcome of Niraparib in a BRCA1-mutant patient with BM from high-grade serious endometrial cancer was highlighted [33] (as reported in Appendix A).

Collectively, these results are consistent with the favorable PK profile of Niraparib in terms of intracranial activity, regardless of BRCA mutation, and its broad clinical activity in intracranial metastases from gynecological malignancies [27,28].

### 3.3. What Evidence Supports Olaparib Maintenance in Patients with Intracranial Relapse?

Case three reflects the clinical setting of Olaparib maintenance in a BRCA-mutated patient after intra-/extracranial recurrence. There is scant evidence supporting Olaparib effectiveness in BM treatment for EOC [14]. We are aware of two case reports showing intracranial responses to Olaparib in EOC/FT/PP cancers, the former with LMs in a BRCA2-mutant patient with HGSOC and the latter with multiple brain metastases in a BRCA1-mutated patient with PP cancer [34,35] (as summarized in Appendix A). Recently, a monocenter case series, dealing with BRCA1-2-mutant patients affected by oligometastatic EOC to the brain, highlighted durable benefit with Olaparib maintenance continued after local therapy (even combined with Bevacizumab) [36]. In another retrospective series of BRCA-mutated patients with BMs, OS benefit from multimodal approaches including PARPi maintenance was reported, although the prognosis remains poor [37].

Interestingly, in a case study, a BRCA1-mutated patient with late isolated CNS relapse yielded a long-term response to Olaparib exceeding 4 years after WBRT completion and 42 months following Olaparib onset. A meaningful intracranial response, with further shrinkage of multiple BMs reported on follow-up MRI, is being maintained along with durable systemic disease control and good QoL, without relevant toxicities [14] (as detailed in the Appendix A). Of note, Olaparib, given as a maintenance regimen in a BRCA2-mutant heavily pretreated patient with platinum-sensitive intracranial relapse, yielded a 14-month disease control intra-and extracranially despite CNS metastases with carcinomatous meningitis [38] (as reported in Appendix A). Patients with LMs harbor a very dismal prognosis [39]. This is the first report highlighting the efficacy of PARPi on meningeal disease of a gBRCA-mut carrier, likely due to the Olaparib ability to cross the leptomeningeal barrier [38].

A peculiar setting is regarding the development of BMs arising from a primary FT cancer, whose incidence is only 0.21% according to the SEER database. In detail, the median time to onset of BMs from FT cancers is about 3 years (range 3–52 months). Of note, sites of CNS metastasis encompass supratentorial and infratentorial compartments as well as the skull base [40]. Among the available reports, a woman diagnosed with a BRCA1-related FT presented with an incidentally discovered left frontoparietal mass, whose pathology was consistent with “metastasis from HGSOC of Mullerian origin”. The adjuvant SRS performed due to the relatively young age, good KPS, and the limited tumor volume yielded sustained benefit. This case is demonstrative of the importance of screening for CNS disease in metastatic gynecologic malignancies, especially in patients harboring BRCA mutations and disseminated disease [40].

### 3.4. What Is the Rationale for Continuing PARPi beyond Intracranial Progression in EOC?

Cases four, five, and six reflect the continuation of PARPi maintenance, both in the frontline (case six) and relapsed (cases four and five) settings, beyond intracranial oligometastatic recurrence without extracranial disease. In detail, they report the Niraparib (cases four–six) and Olaparib (case five) continuation beyond intracranial-only relapse (oligometastases) managed with local therapy. Despite the strong prognostic impact of PARPi in platinum-sensitive recurrence, many patients will eventually progress on maintenance therapy [41]. In the last few years, concerns about post-PARPi progression have emerged, highlighting an unmet need with no validated algorithm strategy [42]. No specific data about BM settings are available. More generally, PARPi may impact both the response to further platinum, due to cross-resistance mechanisms, and subsequent non-platinum CHT as well as surgery [42]. PARPi rechallenge after oligo-progression can be retained as a potential de-escalation strategy in the relapse setting, if combined with locoregional therapies (surgery, ablation, RT), for selected patients [42]. Two retrospective studies highlighted the prolonged clinical benefit from PARPi continuation beyond oligometastatic progression, along with any locoregional therapy, based on the biological rationale of the removal of PARPi-resistant clones in the context of stable disease under PARPi [42].

Compelling evidence from the updated analysis of the phase III ENGOT-OV16/NOVA trial highlights the extended benefit (primary endpoint PFS) of Niraparib monotherapy beyond first disease progression in both the gBRCA-mut and BRCA wild-type cohorts [43]. The final data support the safe long-term use of Niraparib in the platinum-sensitive relapsed setting [41]. However, none of the prospective registration trials addressed PARPi efficacy in BM patients, albeit those with stable metastases being eligible [15]. Supporting evidence from case reports show prolonged intracranial responses with PARPi [15] (as reported in Appendix A). In a recent mono-institutional experience, about one-third of patients on Olaparib maintenance experienced oligo-progression, defined as limited to ≤3 sites, involving CNS at a 5% rate. These patients may benefit from local consolidation therapy, albeit being rarely employed in EOC. No survival differences in patients with and without oligo-progression were found [41] (as detailed in Appendix A). Prospective validation of these findings is mandated to address the value of local therapy for these patients [41].

Strikingly, case six is the first clinical report on the continuation of frontline Niraparib maintenance beyond CNS oligometastatic recurrence. There are no available clinical trials or other case studies regarding this context, probably due to the recent implementation of Niraparib in the frontline setting and the known rarity of CNS relapse. In fact, only one clinical case highlighted the use of frontline Niraparib maintenance in a patient presenting with EOC-related BM at time of initial diagnosis, with no progressive disease [10] (as detailed in Appendix A). Hence, albeit considering the limitations associated with our descriptive case study, especially the low generalizability of these findings, in the absence of high-quality supporting evidence, our experience may add to routine practice for this highly rare scenario and foster further research.

Overall, the indirect encouraging results from trials designed in other tumors, such as non-small cell lung cancer receiving targeted therapies following local ablation, may add to the potential clinical utility of PARPi in intracranial disease control even after oligoprogression [15]. Further consideration should be given to continuing PARPi beyond localized disease control in extracranial oligoprogression [15].

Collectively, the evidence directly ensuring the continuation or not of PARPi after locoregional therapies for CNS progression in EOC is lacking as of yet [15]. Expectedly, it could be argued that the prolonged benefit from PARPi was due to local therapy instead of PARPi. In future, confirmatory data about the role of RT as a valid option for the first oligometastatic platinum-sensitive relapse may allow for the prolongation of the PARPi therapeutic effect beyond oligoprogression [42]. Therefore, large-scale prospective validation in clinical and translational studies is needed to address the role of local therapy for this patient subset [41,42].

## 4. Concluding Remarks and Future Challenges

Owing to the prolonged survival times, uncommon presentations like BMs from EOC are expected to increase. Shared multidisciplinary and supportive care pathways should be ensured for a personalized treatment plan. The gynecologic oncology community will require growing experience with PARPi use in this rare scenario [15]. To our knowledge, EOC patients harboring untreated or symptomatic BMs were excluded from known registration trials [15], and no trials are underway [15]. The continuation of PARPi beyond an oligometastatic progression, including a low-volume CNS disease, has recently emerged as a clinically relevant issue, also due to the potential role of a concomitant local therapy on the prolongation of the PARPi therapeutic effect [15]. The emerging concept of targeted therapy de-escalation could be a new frontier in the personalization of care for EOC. The future challenge is to address, in a timely manner, the dynamic nature and heterogeneity of EOC biology, to find the right personalized therapy for each patient [42]. Until the clinical validation of data about PARPi resistance, enrollment in biomarker-driven clinical trials, testing potential targets for post-progression PARPi combination strategies, should be eagerly encouraged [15,16,17,18,19,20,21,22,23,24,25,26,27,28,29,30,31,32,33,34,35,36,37,38,39,40,41,42].

## Figures and Tables

**Figure 1 ijms-25-07887-f001:**
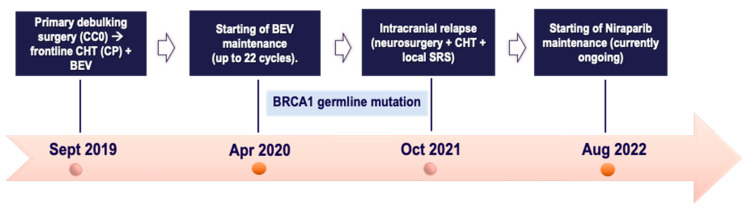
Timeline of patient treatment journey in *case one*. Abbreviations. CC0: completeness of cytoreduction score 0. CHT: chemotherapy. CP: Carboplatin plus Paclitaxel. BEV: Bevacizumab. SRS: stereotactic radiosurgery. Sept: September. Apr: April. Oct: October. Aug: August.

**Figure 2 ijms-25-07887-f002:**
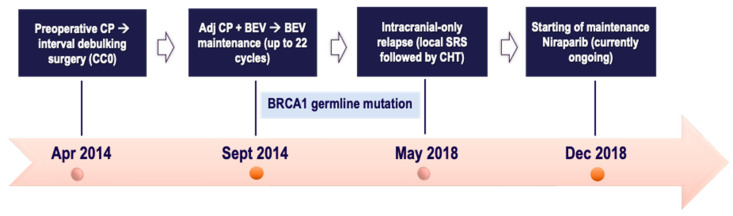
Timeline of patient treatment journey in *case two*. Abbreviations. CC0: completeness of cytoreduction score 0. CP: Carboplatin plus Paclitaxel. CHT: chemotherapy. BEV: Bevacizumab. SRS: stereotactic radiosurgery. Apr: April. Sept: September. Dec: December.

**Figure 3 ijms-25-07887-f003:**
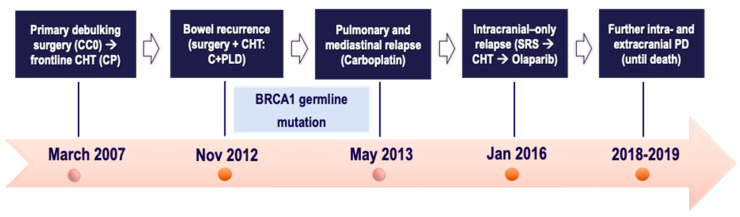
Timeline of patient treatment journey in *case three*. Abbreviations. CC0: completeness of cytoreduction score 0. CHT: chemotherapy. CP: Carboplatin plus Paclitaxel. C + PLD: Carboplatin + Pegylated liposomal doxorubicin. SRS: stereotactic radiosurgery. PD: progressive disease. Nov: November. Jan: January. Nov: November. Jan: January.

**Figure 4 ijms-25-07887-f004:**
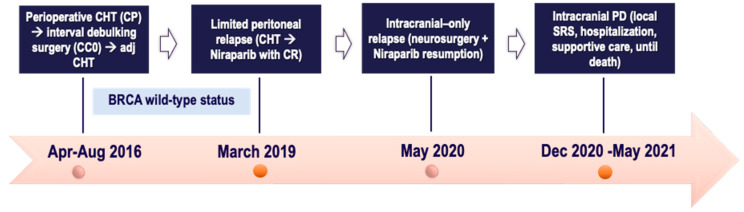
Timeline of patient treatment journey in *case four*. Abbreviations. CC0: completeness of cytoreduction score 0. CHT: chemotherapy. CP: Carboplatin plus Paclitaxel. CR: complete response. PD: progressive disease. SRS: stereotactic radiosurgery. Apr: April. Aug: August. Dec: December.

**Figure 5 ijms-25-07887-f005:**
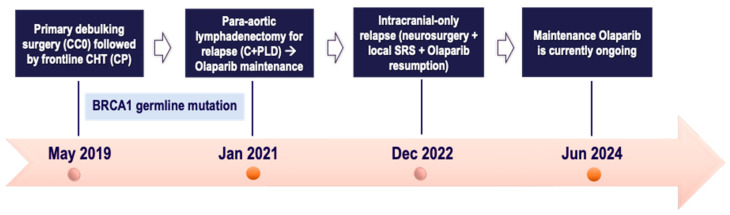
Timeline of patient treatment journey in *case five*. Abbreviations. CC0: completeness of cytoreduction score 0. CHT: chemotherapy. CP: Carboplatin plus Paclitaxel. C + PLD: Carboplatin + Pegylated liposomal doxorubicin. SRS: stereotactic radiosurgery. Jan: January. Dec: December. Jun: June.

**Figure 6 ijms-25-07887-f006:**
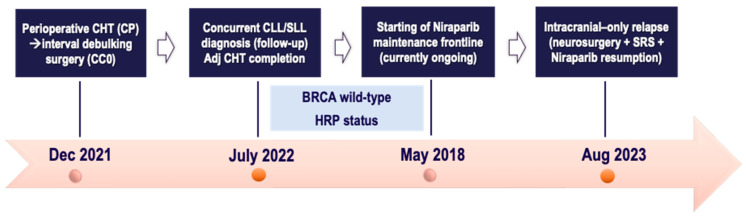
Timeline of patient treatment journey in *case six*. Abbreviations. CC0: completeness of cytoreduction score 0. CHT: chemotherapy. CP: Carboplatin plus Paclitaxel. CLL/SLL: chronic lymphocytic leukemia/small lymphocytic lymphoma. HRP: homologous recombination proficient. SRS: stereotactic radiosurgery. Dec: December. Aug: August.

**Table 1 ijms-25-07887-t001:** Summary of baseline characteristics and treatment outcomes of the patients included in our case study research with BMs from EOC.

Case ID	Age	BRCA Mutation Status	CNS Site (s)	PARPi Agent and Duration	Platinum-Sensitivity Status at Time of BMs	Extracranial Site (s) at Time of BMs	Local Therapies for BMs	CNS BOR and Survival Time with BMs
1	53	BRCA1 PV	Single (cerebellar) BM	Niraparib 18 months (ongoing)	PSR	Para-aortic lymph node	Surgery RT (SRS)	CR 32 months (alive)
2	52	BRCA1 PV	Single (right parietal) BM	Niraparib 64 months (ongoing)	PSR	No	RT (SRS)	CR 72 months (alive)
3	47	BRCA1 PV	Multiple (parietal, occipital) BMs	Olaparib 21 months	PSR (beyond CNS oligo-recurrence)	No	RT (SRS)	PR 46 months (dead)
4	65	BRCA1-2 wild-type	Single (cerebellar) BM	Niraparib 18 months	PSR (beyond CNS oligo-recurrence)	No	Surgery RT (SRS)	PR 12 months (dead)
5	52	BRCA1 PV	Single (single left parieto-occipital) BM	Olaparib 30 months (ongoing)	PSR (beyond CNS oligo-recurrence)	No	Surgery RT (SRS)	CR 18 months (alive)
6	73	BRCA1-2 wild-type	Single (cerebellar) BM	Niraparib 18 months (ongoing)	PSR	No	Surgery RT (SRS)	CR 10 months (alive)

Abbreviations. PV: pathogenic variant. CNS: central nervous system. PSR: platinum-sensitive relapse. BMs: brain metastases. BOR: best overall response. CR: complete response. PR: partial response.

## Data Availability

All relevant data are within the paper and its Appendix A.

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
