# Peer review of "PARP Inhibitors in Brain Metastases from Epithelial Ovarian Cancer through a Multimodal Patient Journey: Case Reports and Literature Review"

_ijms, 2024, doi:10.3390/ijms25147887_

Round 1

Reviewer 1 Report

Comments and Suggestions for Authors

(1) Disease progression during maintenance treatment indicated the failure of the maintenance treatment. In such condition, continuation of that maintenance treatment is against the practice guideline. For case 4, case 5 and case 6, their brain metastasis occurred during their maintenance olaparib/niraparib. This indicated that their maintenance PARPi treatment were not effective. Therefore, the subsequent BMs control of case 5 and case 6 might probably not due to PARPi. This should be clarified in the discussion.

(2) In table 1, “PARPi setting at time of BMs” was in fact “platinum-sensitivity status at time of BMs”.

(3) In table 1, “Extracranial site(s)” should be better clarified as “Extracranial site(s) at the time of BMs”

(4) In table 1, there was “PD” in the abbreviations, but there were only “PR” in the table. Was there anything wrong?

(5) In table 1, for case 3, was her “PSR” also “PSR (beyond CNS oligometastasis)”?

(6) In table 1, for case 3, her lung and mediastinal metastasis occurred before her brain metastasis, not with her brain metastasis.

(7) In table 1, for case 6, her “PSR (beyond CNS oligometastasis)” should be “PSR”.

(8) For case 1, 2, and 3, their BRCA mutation status had been known earlier, but their PARPi maintenance therapies were not started until there later recurrences were controlled. Why?

(9) The reference 3 can be updated to NCCN 2024 version 1.

(10) In line 11 of page 2, “CHT” should be explained (i.e., chemotherapy)

(11) In line 7 of page 3, “[24 Ratner]”  should be corrected.

(12) Case 2 had surgery for brain metastasis in table 2, but not in the text and in figure 2. Which was right?

(13) In figure 2, there was no “CP”.

(14) In line 6 of page 6, “FT” should be explained. Her pathology was HGSOC.

(15) In the 3rd paragraph of page 7, “LM” should be explained.

(16) In the 3rd paragraph of page 7,  was “cinesthesia” in fact “kinaesthesia”?

(17) In the 4th paragraph of page 7, “61 months after diagnosis” was in fact “12 months after diagnosis of brain metastasis”.

(18) For case 6, was her CLL/SLL treated?

(19) In paragraph 2 of page 11, “Cases one and two are emblematic for Niraparib maintenance in patients with intra-cranial-only recurrence” . However, case one also had paraaortic metastasis very near  (2 months) the time of BMs.

(20) In paragraph 2 of page 11, “Cases one and two are emblematic for Niraparib maintenance in patients with intracranial-only recurrence, presenting with a single cerebellar lesion”. However, case two had right parietal metastasis, not cerebellar metastasis.

Author Response

Dear Reviewer, we're very thankful to you and the Editorial Board for having considered our manuscript. We’ve highly appreciated your comments and suggestions as a great opportunity to improve our work, by emphasizing some crucial points. We've thus provided all the point-to-point responses to the comments and suggestions, that are detailed in the attached files. 

Reviewer 2 Report

Comments and Suggestions for Authors

The Authors report six cases of EOC with brain metastases (BM) discussing the correlation between BRCA mutation and this site of disease, the rationale of PARPi, the maintenance with Niraparib and Olaparib. The six clinical cases collected in different Cancer Centers are well presented and the clinical options elegantly discussed.

I have some questions for the Authors:

a) the percentage of BRCA mutations in patients with BM is higher than that observed in high grade serous carcinoma patients. The question is : are BRCAm patients at higher risk of developing BMs or is this observation related to prolonged survival of these patients? In other terms the longer survival correlates with higher risk of BMs? This could be discussed.

b) The surgical treatment of BMs could prolong survival? This approach could be indicated even in presence of extracranial disease?

c) The use of PARPi before the diagnosis of BM could generate chemoresistance mainli in BRCAm tumors?

d) Have different PARPis the same brain penetration or not?

At the end and observation. Probably the paper is too long and could be more readible in a shorter version 

Author Response

(The authors gave the same response as above.)
